# Systematic screening of viral and human genetic variation identifies antiretroviral resistance and immune escape link

Huyen Nguyen[1,2]*, Christian Wandell Thorball[3,4], Jacques Fellay[3,4], Jürg Böni[2], Sabine Yerly[5], Matthieu Perreau[6], Hans H Hirsch[7,8,9], Katharina Kusejko[1,2], Maria Christine Thurnheer[10], Manuel Battegay[8], Matthias Cavassini[11], Christian R Kahlert[12], Enos Bernasconi[13], Huldrych F Günthard[1,2], Roger D Kouyos[1,2]*, The Swiss HIV Cohort Study

[1]Division of Infectious Diseases and Hospital Epidemiology, University Hospital Zurich, University of Zurich, Zurich, Switzerland; [2]Institute of Medical Virology, Swiss National Center for Retroviruses, University of Zurich, Zurich, Switzerland; [3]School of Life Sciences, École Polytechnique, Fédérale de Lausanne, Switzerland; [4]Precision Medicine Unit, Lausanne University Hospital and University of Lausanne, Lausanne, Switzerland; [5]Laboratory of Virology, Geneva University Hospital, University of Geneva, Geneva, Switzerland; [6]Division of Immunology and Allergy, University Hospital Lausanne, University of Lausanne, Lausanne, Switzerland; [7]Transplantation & Clinical Virology, Department of Biomedicine, University of Basel, Basel, Switzerland; [8]Infectious Diseases and Hospital Epidemiology, Department of Medicine, University Hospital Basel, Basel, Switzerland; [9]Clinical Virology, Laboratory Medicine, University Hospital Basel, Basel, Switzerland; [10]University Clinic of Infectious Diseases, University Hospital of Bern, University of Bern, Bern, Switzerland; [11]Department of Infectious Diseases, Centre Hospitalier Universitaire Vaudois, University of Lausanne, Lausanne, Switzerland; [12]Division of Infectious Diseases and Hospital Epidemiology, Kantonsspital St. Gallen, St. Gallen, Switzerland; [13]Division of Infectious Diseases, Regional Hospital, Lugano, Switzerland

*For correspondence:
Huyen.Nguyen@usz.ch (HN);
roger.kouyos@usz.ch (RDK)

## Abstract

**Background:** Considering the remaining threat of drug-resistantmutations (DRMs) to antiretroviral treatment (ART) efficacy, we investigated how the selective pressure of human leukocyte antigen (HLA)-restricted cytotoxic T lymphocytes drives certain DRMs' emergence and retention.

**Methods:** We systematically screened DRM:HLA class I allele combinations in 3997 ART-naïve Swiss HIV Cohort Study (SHCS) patients. For each pair, a logistic regression model preliminarily tested for an association with the DRM as the outcome. The three HLA:DRM pairs remaining after multiple testing adjustment were analyzed in three ways: cross-sectional logistic regression models to determine any HLA/infection time interaction, survival analyses to examine if HLA type correlated with developing specific DRMs, and via NetMHCpan to find epitope binding evidence of immune escape.

**Results:** Only one pair, RT-E138:HLA-B18, exhibited a significant interaction between infection duration and HLA. The survival analyses predicted two pairs with an increased hazard of developing DRMs: RT-E138:HLA-B18 and RT-V179:HLA-B35. RT-E138:HLA-B18 exhibited the greatest significance in both analyses (interaction term odds ratio [OR] 1.169 [95% confidence

interval (CI) 1.075–1.273]; p-value<0.001; survival hazard ratio 12.211 [95% CI 3.523–42.318]; p-value<0.001). The same two pairs were also predicted by netMHCpan to have epitopic binding.

**Conclusions:** We identified DRM:HLA pairs where HLA presence is associated with the presence or emergence of the DRM, indicating that the selective pressure for these mutations alternates direction depending on the presence of these HLA alleles.

**Funding:** Funded by the Swiss National Science Foundation within the framework of the SHCS, and the University of Zurich, University Research Priority Program: Evolution in Action: From Genomes Ecosystems, in Switzerland.

## Introduction

Antiretroviral resistance remains a major obstacle to the successful and lasting suppression of HIV (*Gupta et al., 2012*; *Günthard et al., 2019*). While in resource-rich settings the availability of novel drug classes and personalized HIV treatment have diminished the challenges associated with antiretroviral resistance, resource-limited settings have experienced a continuous increase in antiretroviral resistance, which is now threatening the unprecedented success of the global rollout of antiretroviral treatment (ART) (*Fund, 2019*; *Hauser et al., 2019*). In the context of this globalization of antiretroviral resistance, it is becoming increasingly important to understand how human and viral genetic variation are affecting the processes generating or limiting antiretroviral resistance (*Aghokeng et al., 2011*; *Lataillade et al., 2010*).

HIV drug-resistant mutations (DRMs) can either be selected in patients on ART experiencing treatment failure (acquired drug resistance, or aDRM) or be transmitted from a patient carrying the resistance mutation to an uninfected individual (transmitted drug resistance, tDRM). As some DRMs have been shown to carry a cost, feeding on the virus fitness and replication capacity, they can revert in the absence of ART. Once the selective pressure favoring those mutations is removed, their frequency within a host continuously decreases at the expense of the wild-type variant, and eventually they become undetectable by standard resistance tests. It has been shown that the time scales on which reversion occurs exhibit a large variation ranging from several months to over 10 years, depending on the fitness cost that in turn is governed by both the type of mutation and the genetic background in which it occurs (*Kühnert et al., 2018*; *Yang et al., 2015*). This canonical perspective based on the evolutionary forces of aDRM and tDRM, and their disappearance from the replicating quasi-species, generally disregards the possibility that antiretroviral- resistant mutations are selected in untreated individuals.

One process that may act against the paradigm of DRM emerging only in treated individuals and reverting in untreated individuals is accidental resistance evolution occurring as a collateral effect of viral immune escape. A well-understood instance of this process is evolutionary escape from binding to human leukocyte antigen (HLA), an extremely diverse gene complex encoding for major histocompatibility complex (MHC) proteins. MHC class I proteins (corresponding to HLA class I) are found on the surface of all nucleated cells, and by presenting antigens from the cell interior to the surface, they allow for binding to cytotoxic CD8 T cells (CTL); thus, MHC class I proteins tag the virally infected cell and can subsequently be eliminated by CTL (*Markov and Pybus, 2015*; *Zinkernagel and Doherty, 1979*). The high mutation rate associated with replicating HIV predisposes to cellular and humoral immune escape, where the viral epitopes are no longer recognized by the mounted immune effectors. For CTL-mediated immune responses, this process of developing escape mutations remains a critical part of HIV pathogenesis (*Leslie et al., 2004*). Conversely, the high variability of encoded MHC alleles and their combinations come into play, as the host HLA alleles change as a consequence of transmission (*Markov and Pybus, 2015*; *Zinkernagel and Doherty, 1979*; *Borghans et al., 2004*). If the viral epitope recognized by MHC-I maps to the viral genome at the same region, this could confer an increased viral fitness leading to mutation persistence or even the emergence of a new DRM in an ART-naïve host (*Gatanaga et al., 2013*). While this phenomenon has been reported for individual HIV mutation:HLA pairs, a systematic assessment of the impact of epitope escape across HIV DRM:HLA pairs in a representative population has not yet been reported.

In this study, we investigated and analyzed the viral and genetic data from ART-naïve patients in the Swiss HIV Cohort Study (SHCS). This is leveraging the unique combination of viral and human

genetic data in the SHCS, with over 20,000 genotypic resistance tests and over 5000 patients with information on HLA-I alleles. This allowed us to systematically screen the cohort for associations between DRM:HLA-I pairs and hence for pairs where escape from HLA-I binding might confer the DRM an evolutionary advantage even in the absence of ART.

# Materials and methods

## Swiss HIV Cohort Study

The SHCS is a prospective multicenter study with continuing enrollment, aiming to include all people living with HIV in Switzerland since 1988. About half of all people living with HIV (PLWH) as notified to the Swiss health authorities are voluntarily participating in the SHCS, and include three-quarters of all PLWH receiving ART in the country (*Schoeni-Affolter et al., 2010*). As of August 2019, the SHCS has a cumulative total of 20,741 patients. Demographic information, mode of HIV transmission, treatment, clinical, and other data are updated every 6 months per standard protocol.

## Drug resistance mutation data

The SHCS Drug Resistance Database contains the HIV sequence data, primarily partial *pol* gene sequences, used to determine the presence of DRMs in the viral genome (*von Wyl et al., 2007*). This data, currently covering 13,798 patients, was obtained from both routine clinical testing and systematic retroactive sequencing of stored plasma samples (*Kletenkov et al., 2017*; *von Wyl et al., 2016*). To reduce the scope of our systematic screening to only HIV mutations relevant to drug resistance (thus reducing the risk of overtesting), we only considered the presence of DRMs as defined by the Stanford Drug Resistance Database (*Rhee et al., 2003*). To avoid confounding by the effect of ART, we only considered sequences in ART-naïve individuals (before ART treatment).

## HLA data

Data on the HLA class I type was available for 6453 SHCS patients. This information was obtained from SNP genotype data, using SNP2HLA with the type 1 Diabetes Genetics Consortium reference panel for HLA imputation techniques on the exome/SNP data (*Jia et al., 2013*; *Szolek et al., 2014*; *Dilthey et al., 2016*). We limited our analyses to that of the HLA class I (HLA-A, -B, and -C) considering the existing literature supporting the role of HLA class I peptides in HIV control (*Leslie et al., 2010*; *Pereyra et al., 2010*). Of these patients with HLA data, 3997 additionally had drug resistance testing data.

## Screening candidate pairs of DRM:HLA-type

Our study aimed to retrieve all DRMs identified in the SHCS as well as the HLA-I types found, to analyze whether or not a specific HLA-I type significantly alters the probability of finding a DRM. As there were a possible 5561 combinations represented in our dataset, it was necessary to reduce these candidate pairs to only those for which our data provided sufficient statistical power to detect an association (*Figure 1*). To do this, we filtered out only the combinations where the number of SHCS patients with the given mutation or HLA type were sufficient to provide a statistical power of 0.8, assuming an odds ratio (OR) of 3. This resulted in 225 pairs, from which 225 logistic regression models were made. For each model, the duration of HIV infection time and the presence/absence of the queried HLA-I type were used as predictors of the outcome – the presence of the resistance mutation in the last available resistance test from a given patient when they were ART-naïve. We then used a Benjamini–Hochberg adjustment to account for multiple testing, considering a false discovery rate of 0.2. We purposefully used a more liberal false discovery rate and OR in the prior steps to avoid erroneously discarding any mutation:HLA pair with a potentially valid association, with the intent of compensating for this with the following three analyses assessing the plausibility of the identified pairs:

1. Testing if the impact of duration of HIV infection on the emergence of DRM of interest depends on HLA type: For each candidate pair identified and systematically filtered out after the initial screening, we created a multivariable logistic regression model, where the outcome is the presence of the mutation in the ART-naïve patients in their earliest available sequence (before the start of ART), with the predictors being the presence of the queried HLA type,

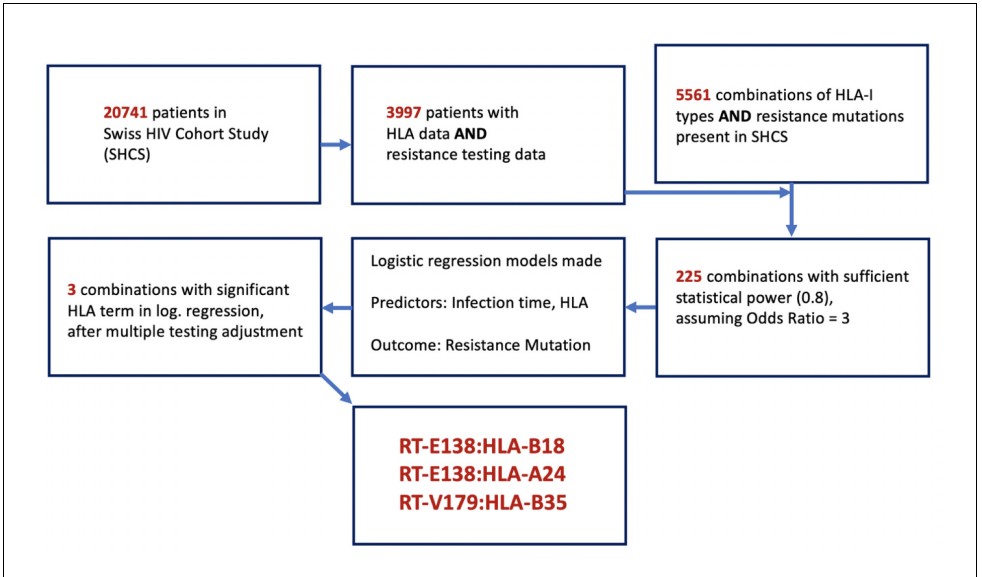

**Figure 1.** Flowchart of methodology of obtaining the candidate DRM:HLA pairs with possible epitope relationship. From the 3997 SHCS patients with both HLA-I data and drug resistance testing data, 5561 potential combinations of HLA-I type and DRMs were examinable, from which only 225 had sufficient power for testing. From these 225, three candidate pairs were found to have a significant HLA term in a logistic regression model predicting the resistance mutation in question. DRM, drug-resistant mutation; HLA, human leukocyte antigen.

duration of HIV infection until time of ART initiation, and additionally, an interaction term between HLA type and infection time. The purpose of the interaction term is to measure if the presence or absence of the queried HLA type affects the selection pressure on the resistance mutation, which would be determined by the interaction term with time since HIV infections – that is, a significant interaction term would imply that time since HIV infection has a different effect on the odds of observing the DRM depending on whether the HLA allele is present or not.

2. Longitudinal/survival analyses: In addition to the cross-sectional logistic regression models, we used Cox proportional hazards survival models to test whether patients initially free of the queried DRM developed it over time. We only considered resistance testing data and time at risk before ART initiation. A patient requires at least two sequences before ART initiation to be included in this analysis. We observed which of the candidate DRM:HLA pairs yielded a survival model where the presence of the queried HLA type was significantly associated with a higher or lower hazard of developing/detecting the mutation over time.

3. Mechanistic plausibility/epitopic binding: To examine whether there was any mechanistic plausibility to the associations found in the above analyses, we utilized the program server NetMHCpan 4.1 to predict the binding affinity of the HLA allele to the all 9-mer peptides including the mutation position, with either the wild-type amino acid at the position or one of the three most common mutated amino acids observed (*Reynisson et al., 2020*). For the candidate pairs where the mutation does cause immune escape, we would anticipate the binding to be stronger for the wild-type peptide compared to the mutated peptides. Additionally, we searched the Los Alamos HIV Molecular Immunology Database to corroborate the candidate pairs with prior experimental studies indicating the HLA–epitope match (*Korber et al., 2021*).

## Software

All analyses (besides the epitope binding predictions performed with netMHCpan) were done in R (version 3.6.1). The code can be found in Github (*Nguyen, 2021*).

## Results

### Obtaining candidate HLA–mutation pairs

From the 20,741 patients in the SHCS, 3997 had both HLA-I alleles data and resistance testing data (*Figure 1*). Characteristics of these patients are shown in *Table 1*. Patients with HLA data were more likely to be Caucasian compared to the general SHCS population, as the HLA SNP imputation methods were validated on a Caucasian population. In the data set, there were 5561 different combinations of HLA-I types represented and DRMs. Only 225 of these pairs had sufficient diversity at the HLA and DRM positions to convey a power greater than 0.8 to detect a strong effect defined as OR = 3 (see 'Materials and methods'). Using logistic regression models, we found three DRM:HLA pairs after multiple testing adjustment (described in *Supplementary file 1*), with a significant impact of the queried HLA type on the odds of observing the DRM: RT-E138:HLA-B18 (OR 6.999, 95% CI 4.662–10.413), RT-E138:HLA-A24 (OR 2.444, 95% CI 1.602–3.658), and RT-V179:HLA-B35 (OR 2.431, 95% CI 1.398–4.108). All three combinations involved a DRM in the reverse transcriptase (RT) gene. Of the three combinations, two were with HLA-B, while one was with HLA-A. These three candidate pairs were further evaluated with three complementary methods: (1) a further cross-sectional analysis examining the presence of an interaction term between infection time and HLA type, (2) a longitudinal survival analysis examining time to the DRM detection among treatment-naïve patients initially without the queried DRM detectable, and (3) NetMHCPan MHC binding prediction analysis to examine mechanistic plausibility.

### HLA-I types and DRMs in study population

The most commonly found HLA-I types are summarized in *Table 2*. Of note, 668 (16.7%) have an HLA-A24 allele, 376 (9.4%) with an HLA-B18 allele, and 728 (18.2%) with HLA-B35. Of the 3997 patients with both DRM and HLA-I data available, 719 (18.0%) had at least 1 DRM, of which 209 (5.2%) had multiple DRMs. Overall, 2267 of all 5155 DRMs in the study population are found among treatment-naïve individuals, and the most frequent of the 1072 DRMs found in the first resistance test in treatment-naïve individuals are summarized in *Table 3*. As for the two DRMs of interest, 145 had a DRM at RT-E138: 124 RT-E138A, 14 RT-E138G, 6 RT-E138K, and 1 RT-E138Q. Eighty-two were found at RT-V179: 68 RT-V179D, 13 RT-V179E, and 1 RT-V179F.

### Cross-sectional analyses/logistic regression models

To examine the effect of having a given HLA-I allele on the presence of the DRM in question, we created for each candidate pair a logistic regression model predicting the presence of that specific DRM (at the earliest resistance testing), given the presence/absence of the queried HLA-I type. From the three candidate pairs, one resultant logistic regression model had a significant interaction term between presence of the queried HLA type and duration of HIV infection (*Figure 2*). For RT-

**Table 1.** General characteristics of SHCS patients and those with resistance mutation and human leukocyte antigen (HLA) data.
Overview of general characteristics of SHCS patients and the subsets with sequencing resistance testing data, HLA-I data, and both. IQR: interquartile range; MSM: men who have sex with men; HET: heterosexual; IDU: intravenous drug use.

|  | All SHCS participants | SHCS patients with resistance testing data | SHCS patients with HLA-I data | SHCS patients with HLA-I and resistance testing data |
|---|---|---|---|---|
| Number | 20,741 | 13,116 | 6450 | 3997 |
| Median age (IQR) | 56 (48–62) | 54 (47–60) | 55 (49–62) | 54 (47–60) |
| Male (%) | 15,064 (72.6%) | 9402 (71.2%) | 4836 (75.0%) | 3027 (75.7%) |
| Risk group: MSM | 8100 (39.1%) | 5226 (39.8%) | 2777 (43.1%) | 1784 (44.6%) |
| HET | 6841 (33.0%) | 4731 (36.1%) | 2173 (33.7%) | 1439 (36.0%) |
| IDU | 4840 (23.3%) | 2568 (19.6%) | 1255 (19.5%) | 620 (15.5%) |
| Other | 960 (4.6%) | 591 (4.5%) | 245 (3.8%) | 154 (3.9%) |
| White (%) | 14044 (67.7%) | 9993 (76.2%) | 5661 (87.8%) | 3487 (87.2%) |

**Table 2.** Distribution of most common HLA-I A, B, and C alleles in study population.
Ten most common HLA-A, -B, and -C types in study population individuals with both HLA-I and DRM information. Frequency and percentage of individuals with each allele are indicated. DRM, drug-resistant mutation; HLA, human leukocyte antigen.

| HLA-A type | Frequency | Percentage |
|---|---|---|
| 02 | 1838 | 46.0 |
| 03 | 964 | 24.1 |
| 01 | 857 | 21.4 |
| 24 | 668 | 16.7 |
| 11 | 493 | 12.3 |
| 68 | 340 | 8.5 |
| 32 | 302 | 7.6 |
| 30 | 300 | 7.5 |
| 26 | 272 | 6.8 |
| 29 | 261 | 6.5 |
| HLA-B type | Frequency | Percentage |
| 44 | 905 | 22.6 |
| 07 | 814 | 20.4 |
| 35 | 729 | 18.2 |
| 51 | 639 | 16.0 |
| 15 | 582 | 14.6 |
| 08 | 500 | 12.5 |
| 40 | 410 | 10.3 |
| 18 | 376 | 9.4 |
| 57 | 328 | 8.2 |
| 27 | 294 | 7.4 |
| HLA-C type | Frequency | Percentage |
| 07 | 1794 | 44.9 |
| 04 | 941 | 23.5 |
| 03 | 812 | 20.3 |
| 06 | 772 | 19.3 |
| 12 | 510 | 12.8 |
| 05 | 485 | 12.1 |
| 02 | 401 | 10.0 |
| 16 | 341 | 8.5 |
| 01 | 328 | 8.2 |
| 15 | 320 | 8.0 |

E138:HLA-B18, duration of HIV infection (OR 0.918, 95% CI 0.862–0.971 [p-value=0.004]) and the HLA:time-to-DRM interaction term (OR 1.169, 95% CI 1.075–1.273 [p-value<0.001]) were both significant predictors of an RT-E138 mutation. Greater infection time was thus correlated with a smaller chance of having/detecting the RT-E138 mutation (due to the fitness cost of the mutation). However, in individuals with HLA-B18, the HLA:time-to-DRM interaction terms cause the selection pressure to reverse direction, hence greater infection time is instead correlated with a greater probability of an RT-E138 mutation for HLA-B18 individuals.

**Table 3.** Distribution of most common drug-resistant mutations (DRMs) in study population.
Ten most common DRMs from the earliest available resistance testing of the study population, with the frequency and percentage of each among the study population indicated. Specific amino acid mutations represented in the population are shown.

| Gene | Specific DRM | Frequency | Percentage |
|---|---|---|---|
| RT-E138 | AGKQ | 145 | 3.63 |
| RT-T215 | ACDEFILNSVY | 132 | 3.30 |
| RT-V106 | AIM | 95 | 2.38 |
| RT-V179 | DEF | 82 | 2.05 |
| RT-M41 | L | 72 | 1.80 |
| PR-M46 | ILV | 47 | 1.18 |
| RT-K103 | NS | 46 | 1.15 |
| RT-K219 | ENQR | 34 | 0.85 |
| RT-D67 | EGN | 34 | 0.85 |
| RT-M184 | IV | 30 | 0.75 |

## Longitudinal/survival analyses

To examine the effect of having a given HLA-I allele on the development of the DRM in question, we performed for each pair a survival analysis to observe how many individuals initially without the DRM eventually develop it prior to initiation of ART. Two of the three candidate DRM:HLA pairs were shown to have a significant difference in the probability of the queried mutation arising in initially wild-type individuals. For RT-E138:HLA-B18, 63 (7.7%) of the 813 patients without an RT-E138 mutation were HLA-B18, among which 5 (7.9%) developed it before ART initiation, compared to the 5 (0.7%) of the 750 with another HLA-B18 type (hazard ratio [HR] 12.211, 95% CI 3.523–42.318 [p-value<0.001]) (*Figure 3*). RT-V179:HLA-B35 showed a similarly sharpened increased hazard of developing the mutation. Of the 150 (18.3%) of the 821 patients with HLA-B35 (initially without an HLA-B35 mutation), 3 (2.0%) developed a mutation at RT-V179, compared to only 1 (0.1%) of the 671 with another HLA-B type (HR 16.116, 95% CI 1.673–155.216 [p-value=0.016]).

## Mechanistic plausibility/epitope binding

NetMHCpan predictions of HLA binding were performed to gauge the mechanistic plausibility of the effects observed in the first two analyses. These also indicated weakened HLA binding to the DRM-peptide (i.e. supporting the putative association) for two of the three candidate pairs: RT-E138:HLA-B18 and RT-V179:HLA-B35 (*Supplementary file 2*). Thus, in these two DRM:HLA pairs, the HLA-I allele is driving viral immune escape by reducing avidity to MHC. The two pairs supported by mechanistic plausibility are the same two pairs having a significant relationship between HLA type presence and survival in the longitudinal analyses (*Table 4*). Prior literature indicating experimentally verified epitope binding of the HIV proteome to HLA also exists for these two pairs (*Gatanaga et al., 2013*; *Kopycinski et al., 2014*; *Liu et al., 2006*; *Li et al., 2011*; *Llano et al., 2019*; *Pereyra et al., 2014*; *Kiepiela et al., 2007*; *Peretz et al., 2011*; *Rowland-Jones et al., 1995*; *Tebit et al., 2009*; *Bond et al., 2001*).

## Discussion

Our analyses indicate strong evidence for the presence of an evolutionary intrapatient interaction between HIV DRMs and certain HLA-I alleles. Of the three candidate DRM:HLA pairs analyzed by three methods, two were supported by two of the analyses to show this relationship, of which one, RT-E138:HLA-B18, was supported by all three (*Table 4*). This is notable as this pair has been specifically investigated by *Gatanaga et al., 2013*, who showed both experimentally and through structural modeling that HLA B18-restricted CTLs select for a mutation in RT138. Our study independently demonstrates that this interaction is relevant at the population level, both in cross-sectional and in longitudinal cohort data. Of note, both DRMs are associated with the nonnucleoside analogue

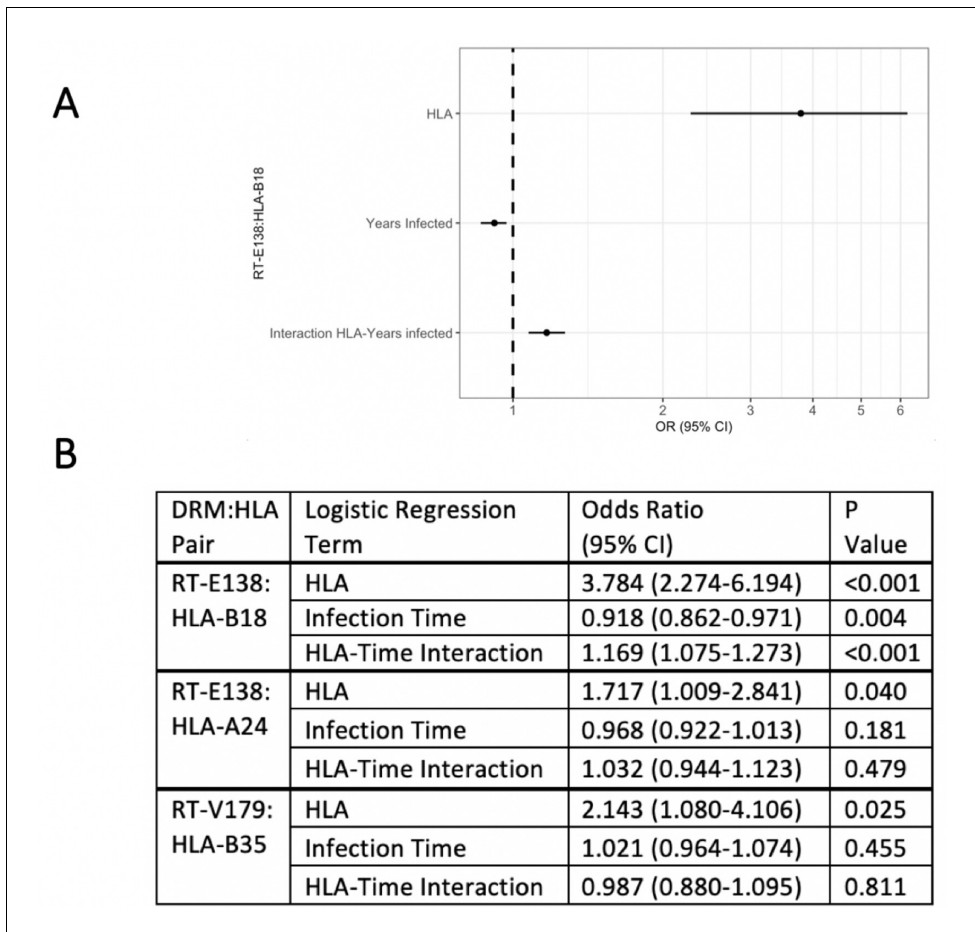

**Figure 2.** Logistic regression models testing for interaction between the queried human leukocyte antigen (HLA) type and duration of infection in predicting the presence of drug-resistant mutation (DRM). Of the three candidate DRM:HLA type pairs, one pair, RT-E138:HLA-B18, indicates a significant interaction term between the presence of the queried HLA type and the duration of HIV infection in a logistic regression model predicting the presence of a mutation at RT-E138 (**A**). (**B**) Details of all three candidates' logistic regression models.

reverse transcriptase inhibitor class of ART drugs, with RT-V179D/F/T being associated with resistance to Etravirine and RT-V179L being associated with Rilpivirine. RT-E138A/G/K/Q is associated with resistance to Etravirine and Rilpivirine (*International Antiviral Society, 2019*). Estimates of virological failure for these two drugs are upwards of 5% and 11%, for Efavirenz and Rilpivirine, respectively (*Sanford, 2012*).

These results have major implications for our understanding of the evolutionary epidemiology in viral infections as they demonstrate a considerable interaction between the processes of drug resistance evolution and immune escape observed for several drug classes and HLA alleles in a representative patient population. This extends the standard paradigm that resistance mutations are acquired in treated individuals, may become transmitted, but eventually disappear in treated individuals with the possibility that resistance mutations newly emerge in untreated individuals due to immune escape. While this mechanism does obviously not account for the majority of DRMs in patients with untreated HIV, it may not be a negligible phenomenon.

In fact, HLA type-driven viral evolution in DRM-relevant CTL epitopes may be particularly relevant in light of the estimated 10% with a DRM in ART-naïve European HIV-positive patients, and even higher figures in low-resource settings, where continuing issues with access to treatment and adherence exacerbate the risk of treatment failure (*Günthard et al., 2019*; *Hofstra et al., 2016*; *Wittkop et al., 2011*; *Chimukangara et al., 2019*; *Pessôa and Sanabani, 2017*). As HLA is extremely diverse in the human population, and exhibits high variation in allelic frequency in

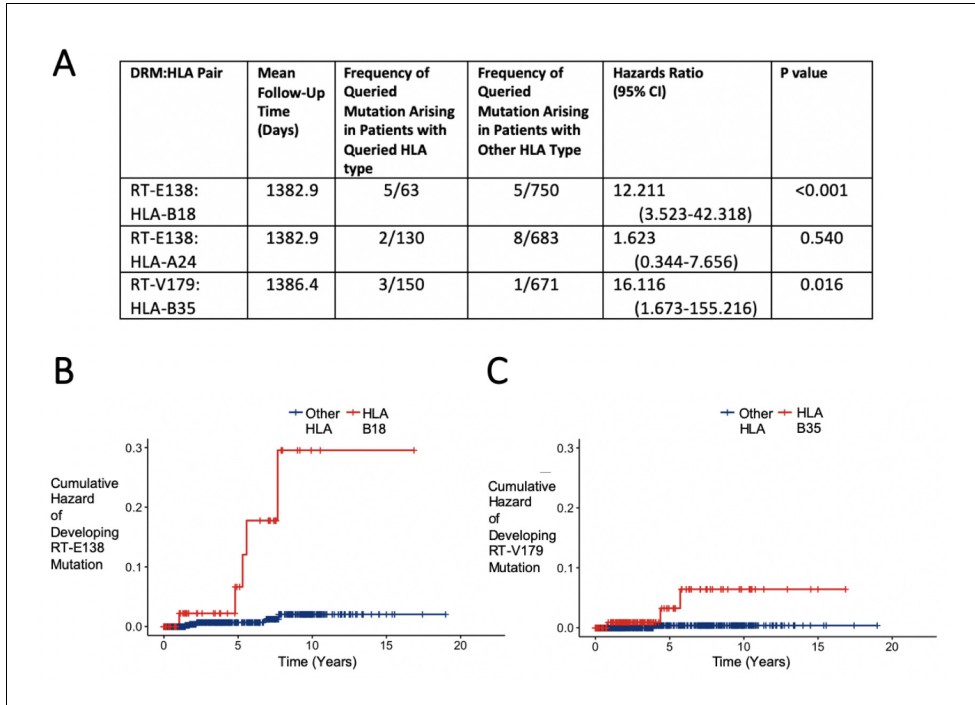

**Figure 3.** Hazard ratios and cumulative hazards of developing queried drug-resistant mutation over time in relation to the presence of human leukocyte antigen (HLA) type. (A) Cox proportional hazard ratios for developing the queried drug-resistant mutation with the queried HLA-I type. (B, C) Cumulative hazard plots of the two pairs from (A) where the hazard ratios were significant, indicating cumulative hazards of developing the mutation among those initially wild type, with red lines indicating individuals with the queried HLA type and blue lines for those with another HLA type.

different geographic regions (*Piazza et al., 1980*), this DRM:HLA link may partially explain regional variations in pre-treatment drug resistance. Accordingly, we would expect the emergence of certain DRMs in the population that is ART-naïve, or specifically, naïve to Etravirine and Rilpivirine, if the local population has a higher prevalence of the HLA types indicated in our analyses.

As the SHCS primarily consists of individuals of white ethnicity from Switzerland and surrounding countries, our study is statistically best powered to detect DRM:HLA pairs amongst white patients, and may be too underpowered to detect DRM:HLA pairs involving HLA-I alleles more prevalent in non-white, low-resource settings – precisely where DRMs are a more urgent issue. This is even concerning considering the high number of pairs eliminated after filtering out those with insufficient numbers to power an analysis (*Figure 1*). This lack of power may explain, for example, RT-V179:HLA-B35 indicates a DRM:HLA association in the longitudinal analysis, but not in the cross-sectional analysis with the interaction term (*Table 4*). It is conceivable that with greater numbers of patients and more years of follow-up that more DRM:HLA pairs would be detected and that these inter-analyses inconsistencies would be resolved, though we should not exclude the possibility of other

**Table 4.** DRM:HLA pairs corroborated by each analytical approach.
Summary of HLA–drug-resistant mutation pairs in all three approaches. Methods that corroborate the HLA–mutation relationship are indicated by 'yes.' DRM, drug-resistant mutation; HLA, human leukocyte antigen.

| DRM:HLA pair | Interaction term in cross-sectional logistic regression | Longitudinal/ survival analysis | Mechanistic plausibility |
|---|---|---|---|
| RT-E138:HLA-B18 | Yes | Yes | Yes |
| RT-E138:HLA-A24 | No | No | No |
| RT-V179:HLA-B35 | No | Yes | Yes |

sources for such discrepancies, for example, imprecise estimates of HIV infection time. The limitation of most sequences to the *pol* gene also made the analyses underpowered to find DRM:HLA relationships in other genes.

Despite these limitations, our study is strengthened by its methodological breadth and thoroughness. While other studies have examined the link between HLA-I and DRMs (*Ahlenstiel et al., 2007*; *Bailey et al., 2007*), this study is on a numerically larger scale, and is unique to systematically examine an entire HIV cohort population's DRM profiles and HLA-I types to screen for potential DRM:HLA pairs. As the cross-sectional analysis took into account duration of infection, it thus effectively excluded from consideration tDRMs that were disadvantageous to viral fitness in ART-naïve patients, identifying any DRMs that remained over time despite the lack of selection pressure from ART, thus mitigating the possibility that these DRMs are merely tDRMs with no relevance to viral pathogenesis in the patient. Additionally, as it is now clinical practice to immediately initiate ART in newly diagnosed patients since several years, there is now hardly ever more than one ART-naïve sequence per patient, thus making our longitudinal analysis very unique and difficult to replicate in the future (*World Health Organization, 2016*; *Ryom et al., 2016*).

By utilizing three different analytical approaches, especially by combining the longitudinal and cross-sectional approaches, we are able to identity and validate DRM:HLA pairs where there is this epitope–mutation interaction. The NetMHCPan analyses allowed us to connect the associations we statistically detected at a population level with predicted MHC binding, which was additionally supported by prior experimental findings. This screening process is also strengthened by the restriction to pairs where the HLA-I and DRM frequencies have sufficient power, thus reducing the number of performed tests and the magnitude of the Benjamini–Hochberg multiple testing adjustment.

Our findings not only have an impact on our understanding of why DRMs tend to be transmitted and maintained in certain individuals, but may also help inform ART in the future. While it would not be feasible to tailor ART treatment based on personal HLA genotyping in resource-limited settings, this information could be used to help anticipate a higher frequency of certain DRMs where a corresponding HLA-I type is more prevalent. As HIV sequencing progresses, more complete DRM:HLA data on other genes, particularly integrase, will become available at sufficiently powered frequencies, enabling us to detect potential DRM:HLA pairs that may affect the efficacy of integrase inhibitors, a newer and increasingly used ART drug class.

## Acknowledgements

The authors thank the patients who participated in the Swiss HIV Cohort Study; the physicians and study nurses, for the excellent patient care provided to participants; the resistance laboratories for high-quality genotyping drug resistance testing; SmartGene (Zug, Switzerland), for technical support; Alexandra Scherrer, Susanne Wild, and Anna Traytel from the SHCS data center for data management; and Marianne Amstutz, Danièle Perraudin, and Mirjam Minichiello for administration. The members of the Swiss HIV Cohort Study include the following: A Anagnostopoulos, MB, EB, JB, D L Braun, H C Bucher, A Calmy, MC, A Ciuffi, G Dollenmaier, M Egger, L Elzi, J Fehr, JF, H Furrer (chairman of the Clinical and Laboratory Committee), C A Fux, H F Günthard (president of the SHCS), D Haerry (deputy of 'Positive Council'), B Hasse, HH Hirsch, M Hoffmann, I Hösli, M Huber, C Kahlert, L Kaiser, O Keiser, TK, RD Kouyos, H Kovari, B Ledergerber, G Martinetti, B Martinez de Tejada, C Marzolini, KJ Metzner, N Müller, D Nicca, PP, G Pantaleo, MP, A Rauch (chairman of the Scientific Board), C Rudin (chairman of the Mother and Child Substudy), K Kusejko (head of Data Center), P Schmid, R Speck, M Stöckle, P Tarr, A Trkola, PV, G Wandeler, R Weber, and SY.

## Additional information

### Competing interests

Matthias Cavassini: has received research and travel grants for his institution from ViiV and Gilead. Enos Bernasconi: has received fees for his institution for participation to advisory board from MSD, Gilead Sciences, ViiV Healthcare, Abbvie and Janssen. Huldrych F Günthard: HFG has received unrestricted research grants from Gilead Sciences and Roche; fees for data and safety monitoring board membership from Merck; consulting/advisory board membership fees from Gilead Sciences, Sandoz

and Mepha; and travel reimbursement from Gilead. The other authors declare that no competing interests exist.

## Funding

| Funder | Grant reference number | Author |
|---|---|---|
| University of Zurich | University Research Priority Program, "Evolution in Action: From Genomes to Ecosystems": U-702-26-01 | Huyen Nguyen Roger D Kouyos |
| Schweizerischer Nationalfonds zur Förderung der Wissenschaftlichen Forschung | BSSGI0_155851 | Huldrych F Günthard |
| Schweizerischer Nationalfonds zur Förderung der Wissenschaftlichen Forschung | 179571 | Huldrych F Günthard |
| Schweizerischer Nationalfonds zur Förderung der Wissenschaftlichen Forschung | 148522 | Huldrych F Günthard |

The funders had no role in study design, data collection and interpretation, or the decision to submit the work for publication.

## Author contributions

Huyen Nguyen, The Swiss HIV Cohort Study, Conceptualization, Software, Formal analysis, Validation, Investigation, Visualization, Methodology, Writing - original draft, Writing - review and editing; Christian Wandell Thorball, Resources, Data curation, Software, Validation, Writing - review and editing; Jacques Fellay, Resources, Data curation, Supervision, Methodology, Writing - review and editing; Jürg Böni, Sabine Yerly, Matthieu Perreau, Hans H Hirsch, Maria Christine Thurnheer, Manuel Battegay, Matthias Cavassini, Christian R Kahlert, Enos Bernasconi, Resources, Data curation, Writing - review and editing; Katharina Kusejko, Software, Supervision, Writing - review and editing; Huldrych F Günthard, Resources, Data curation, Supervision, Funding acquisition, Methodology, Project administration, Writing - review and editing; Roger D Kouyos, Conceptualization, Software, Supervision, Funding acquisition, Methodology, Project administration, Writing - review and editing

## Author ORCIDs

Huyen Nguyen [ID] https://orcid.org/0000-0003-1486-8970
Jacques Fellay [ID] http://orcid.org/0000-0002-8240-939X
Katharina Kusejko [ID] http://orcid.org/0000-0002-4638-1940
Christian R Kahlert [ID] https://orcid.org/0000-0002-0784-3276
Roger D Kouyos [ID] http://orcid.org/0000-0002-9220-8348

## Ethics

Human subjects: The SHCS has been approved by the participating institutions' ethics committees (Kantonale Ethikkommission Bern, Ethikkommission des Kantons St. Gallen, Comité; Départemental dÉthique des Spécialités Médicales et de Médecine Communautaire et de Premier Recours, Kantonale Ethikkommission Zurich, Repubblica e Cantone Ticino-Comitato Etico Cantonale, Commission Cantonale d'Éthique de la Recherche sur l'tre Humain, Ethikkommission beider Basel; all approvals are available at http://www.shcs.ch/206-%0Dethic-committee-approval-and-informed-consent). Written informed consent was obtained from all participants.

## Decision letter and Author response

Decision letter https://doi.org/10.7554/eLife.67388.sa1
Author response https://doi.org/10.7554/eLife.67388.sa2

## Additional files

### Supplementary files

• Source data 1. Data files with select anonymized variables necessary for producing main figures.

• Supplementary file 1. Overview of Benjamini–Hochberg adjustment of DRM:HLA candidate pairs. Overview of the Benjamini–Hochberg procedure to correct for multiple testing in selecting HLA–mutation pairs. Pairs were ranked by the p-value of the HLA term in the adjusted logistic regression model predicting for the queried mutation. The numerical rank (I) is divided by the total number of pairs (m = 225) and multiplied by the false discovery rate of 0.2 (Q). With this adjustment, the lowest-ranked pairs where the p-value is lower than (I/m)Q, along with all pairs ranked above, are included after the adjustment (in bold in the table), yielding the three candidate pairs we investigated in-depth (in bold). Only the first 25 rows of the total 225 are shown.

• Supplementary file 2. Table of NetMHCpan predictions of top binding peptides for each HLA–DRM candidate pair. For each HLA–mutation pair, the binding peptides (defined as below a rank of 2% for weakly binding and below 0.5% for strongly binding) are listed ranked in decreasing predicted binding strength according to NetMHCpan. Peptides in bold denote the peptides without the mutation that bind more strongly than all other peptides for that position in the viral amino acid sequence. Peptides in bold and italics denote peptides without the mutation that bind more weakly than a mutated form.

• Transparent reporting form

### Data availability

The individual-level datasets generated and analyzed for the current study do not fullfill the requirements for open data access: (1) The SHCS informed consent states that sharing data outside the SHCS network is only permitted for specific studies on HIV infection and its related complications, and to researchers who have signed an agreement detailing the use of the data and biological samples; and (2) the data is too dense and comprehensive to preserve patient privacy in persons living with HIV. Per Swiss law, data cannot be shared if data subjects have not agreed or if data is too sensitive to share. Investigators with a request for the data that support the findings of this study should contact the corresponding author Roger Kouyos and the Scientific Board of the SHCS. The provision of data will be considered by the Scientific Board of the SHCS and the study team and is subject to Swiss legal and ethical regulations, and is outlined in a material and data transfer agreement. We have however, provided the data files (with the rows anonymized and randomly re-assorted) with the bare minimum number of variables necessary to do the core analyses and to assemble the figures as shown in the manuscript. The code for the analysis can be found on Github repository: https://github.com/hnyhnyhny/HNGUYEN_HLA_DRM (copy archived at https://archive.softwareheritage.org/swh:1:rev:4a03919f07748ff22c4bf529100505ecc78b57cd).

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
