## [Decision Letter]

**Acceptance summary:**

This paper uses a rigorous methodological approach to identify an interesting and novel mechanism of HIV drug resistance in addition to acquired and transmitted drug resistance: accidental resistance arising coincidentally due to immune escape. This finding is of potential clinical importance as such mutations can arise spontaneously during untreated HIV infection in the context of a specific HLA-type.

**Decision letter after peer review:**

Thank you for submitting your article "Systematic screening of viral and human genetic variation identifies antiretroviral resistance and immune escape link" for consideration by *eLife*. Your article has been reviewed by 2 peer reviewers, including Joshua T Schiffer as the Reviewing Editor and Reviewr #1, and the evaluation has been overseen by Jos van der Meer as the Senior Editor.

Essential revisions:

1. Please expand and clarify the description of the DRM and genotyping data. Please specify the number of different HLA-types, the relative frequencies of the predominant HLA-types, the number of different drug resistance variants found in ART-naïve patients and their relative frequency in the population as well as frequencies of multiple DRMs.

2. In the results, please include a first paragraph that puts the subsequent analysis into context and describe the purpose of the cross-sectional and survival analyses before presenting the results.

3. Either eliminate the term "literature search" or specifically describe the search methodology that was used to perform this search.

4. Include viral load data if available as described by Reviewer 1.

*Reviewer #1:*

This paper by Nguyen and colleagues identifies 3 possible HIV mutations which are linked by HLA-association and drug resistance. The conclusions are justified based on the analysis. The results are unlikely to influence clinical practice as the identified mutations are of only occasional importance in the clinic. However, the general concept is of great relevance to HIV and other viral infections.

Strengths of the study are:

1. Novel conclusions: the paper is important because it identifies a new source of possible drug resistance in addition to selective pressure during failed therapy and transmitted drug resistance.

2. Methodology strength: the authors do a nice job of attempting to move towards causality rather than correlation. Specifically, they establish strength of correlation, temporality (the survival analysis showing accrual of the new mutation over time), assessment of possible confounding variables such as ruling out transmitted mutations with the survival analysis and looking at an interaction term with infection time, assessing mechanistic plausibility with literature review and utilizing a very large cohort.

3. Thoughtful and clearly written intro and discussion.

There were no major weaknesses with the study from my perspective.

This study was extremely clearly written and I believe the scientific conclusions are justified by the analysis. One area of interest would be viral loads among study participants with specific HLA/drug resistance pairs. If possible and if data is available, then it would be interesting to see whether onset of a new mutation as seen in the survival analysis is associated with a decrease in viral load due to a fitness cost. There would be a natural comparison to make versus viral load changes in all other participants, as well as those with relevant HLA types who do not develop new drug resistance mutations. A similar approach with CD4 T cell count trajectories would also be interesting and in theory easy to perform.

*Reviewer #2:*

Drug resistance mutations (DRMs) are known to emerge and be selected for in patients on ART experiencing treatment failure. Transmission of DRMs can lead to high levels of drug resistance in treatment-naïve individuals in resource-limited settings, which may compromise first-line treatment regimens. Here the authors investigated whether the emergence and persistence of DRMs in untreated individuals could also occur as a side effect of immune selection. To this end, they screened for associations between drug resistance mutations and different HLA-1 alleles in a large cohort of nearly 4,000 treatment-naïve patients.

The authors identified three DRM-HLA-1 pairs where the presence of the HLA-1 allele was predictive of the patient having acquired the DRM prior to starting ART. Similar analyses of other patients cohorts have also identified potential associations between DRMs and specific HLA-1 alleles, so in that sense, the novelty of these findings is limited. However, this study substantially extends previous work by incorporating additional analyses that account for the duration of infection before treatment and the time to emergence of drug resistance, leveraging their exceptionally detailed dataset. These analyses revealed that patients without the HLA-B18 allele who had acquired the DRM at transmission were likely to revert to wildtype over time due to the fitness cost associated with the mutation, whereas patients with the HLA-B18 allele either retained the transmitted DRM or generated it de novo to escape immune pressure. Longitudinal survival analysis further indicated that patients with the HLA-B18 and HLA-B35 alleles who did not initially have the associated DRMs were significantly more likely to acquire them than patients without the alleles.

Overall, the conclusions of this paper are well supported by data. The statistical methods used are rigorous and straightforward to apply to cohort data for other populations. However, the description of the DRM and genotyping data is minimal, and its presentation is somewhat confusing. While the three identified HLA-DRM pairs were analyzed extensively, it is not clear how frequent these genotypes and DRMs (independently of each other) are in the patient cohort. As currently written, the manuscript lacks context for assessing the contribution of immune-driven DRMs relative to transmission-acquired DRMs in the treatment-naïve HIV+ population. Overall, however, this paper provides a valuable contribution to the field as a proof-of-concept demonstration of the emergence of immune-derived DRMs.

The Results section would greatly benefit from a first paragraph that puts the subsequent analysis into context. Currently it is challenging to get any sense for the distribution of DMRs and HLA-types in the population. It would help to specify the number of different HLA-types, the relative frequencies of the predominant HLA-types, the number of different drug resistance variants found in ART-naïve patients and their relative frequency in the population. What proportion of the patients that were included in the study had any drug resistance mutations at all? Did any patients have multiple DRMs?

In my opinion, the Results section would flow more logically if the purpose of the cross-sectional and survival analyses were discussed before presenting the results. E.g. A cross-sectional analysis was performed to assess "…" and a survival analysis to assess "…".

Specifying "literature search" as an analysis methodology seems a little clunky. Why not simply state that the B18 HLA type has been shown to bind to the epitope containing the E138 DR mutation in several previous studies [listing citations]?

Would the cross-section analysis have more power if the two potential HLA alleles associated with the PR-E138 DRM were compared simultaneously against a reference group containing all other alleles? In the current analysis, the two potentially important alleles are essentially compared against each other (as part of the "other" category).

It would be helpful if the frequencies in Figure 3A would be spelled out in text rather than just shown in the table. E.g. what proportion of patients had the DRM-associated HLA alleles, and what fraction of these patients subsequently acquired the DRMs?

---

## [Author Response]

Reviewer #1:[…] This study was extremely clearly written and I believe the scientific conclusions are justified by the analysis. One area of interest would be viral loads among study participants with specific HLA/drug resistance pairs. If possible and if data is available, then it would be interesting to see whether onset of a new mutation as seen in the survival analysis is associated with a decrease in viral load due to a fitness cost. There would be a natural comparison to make versus viral load changes in all other participants, as well as those with relevant HLA types who do not develop new drug resistance mutations. A similar approach with CD4 T cell count trajectories would also be interesting and in theory easy to perform.

While this is a very relevant question and would undoubtedly add to the study, we are unfortunately limited by power. For 2 of the pairs: RT-E138:HLA-A24 and RT-V179:HLA-B35, there are 2 and 3 mutation events respectively among those with the queried HLA type, and 8 and 1 respectively for those with another HLA type. This makes a paired T-test comparing the mean HIV viral load or CD4 count before and after the mutation nearly impossible to detect any difference, even if there is one.

To illustrate this, we did perform the comparison for RT-E138:HLA-B18:

The average viral load right after the mutation for those with HLA-B18 was 51 489, compared to the much higher 97 098 viral load mean before the mutation arose. Although the viral load is apparently halved, the fact that this was pooled from the four out of the five individuals with a post and pre-mutation viral load measurement, meant that the p value of the paired T-test (after log-transforming the viral load values), was only 0.122. While this indicates a probable relationship (particularly in contrast to the increase among the 5 non-HLA-B18 individuals who developed the mutation [mean viral load before: 23 036, after: 35 756, p value=0.137), we would need more patients to definitively demonstrate this. There was no change observed for CD4 count.

Because of this, we regretfully decided not to include this into the manuscript.

Reviewer #2:[…] The Results section would greatly benefit from a first paragraph that puts the subsequent analysis into context. Currently it is challenging to get any sense for the distribution of DRMs and HLA-types in the population. It would help to specify the number of different HLA-types, the relative frequencies of the predominant HLA-types, the number of different drug resistance variants found in ART-naïve patients and their relative frequency in the population. What proportion of the patients that were included in the study had any drug resistance mutations at all? Did any patients have multiple DRMs?

This is a very valid point. Two tables have now been added to indicate the most frequent HLA-I alleles and DRMs observed in the study population (Tables 2-3, pg 14-15). A new subsection (lines 270-280) has been added to the Results to summarize them, before moving on to the analyses themselves (the DRM breakdown originally in the Discussion has also been moved to this subsection):

“The most commonly found HLA-I types are summarized in Table 2. […] As for the two DRMs of interest, 145had a DRM at RT-E138: 124 RT-E138A, 14 RT-E138G, 6 RT-E138K, 1 RT-E138Q. Eighty-two were found at RT-V179: 68 RT-V179D, 13 RT-V179E, 1 RT-V179F.”

In my opinion, the Results section would flow more logically if the purpose of the cross-sectional and survival analyses were discussed before presenting the results. E.g. A cross-sectional analysis was performed to assess "…" and a survival analysis to assess "…".

An introduction to each analysis has been added to the beginning of each analysis subsection. (lines 294-297, 319-321, 345-346):

“To examine the effect of having a given HLA-I allele on the presence of the DRM in question, we created for each candidate pair a logistic regression model predicting the presence of that specific DRM (at the earliest resistance testing), given presence/absence of the queried HLA-I type. […] These also indicated weakened HLA binding to the DRM-peptide (i.e. supporting the putative association) for two of the three candidate pairs: RT-E138:HLA-B18 and RT-V179:HLA-B35 (Supplementary Table S2).”

Specifying "literature search" as an analysis methodology seems a little clunky. Why not simply state that the B18 HLA type has been shown to bind to the epitope containing the E138 DR mutation in several previous studies [listing citations]?

Methods:

“To examine if there was any mechanistic plausibility to the associations found in the above analyses, we utilized the program server NetMHCpan 4.1 to predict the binding affinity of the HLA allele to the all 9-mer peptides including the mutation position, with either the wildtype amino acid at the position or one of the three most common mutated amino acids observed (24). […] Additionally, we searched the Los Alamos HIV Molecular Immunology Database to corroborate the candidate pairs with prior experimental studies indicating the HLA-epitope match (25).”

Results:

“NetMHCpan predictions of HLA binding were performed to gauge the mechanistic plausibility of the effects observed in the first two analyses. […] Prior literature indicating experimentally verified epitope binding of the HIV proteome to HLA also exists for these two pairs (13, 27, 28, 29, 30, 31, 32, 33, 34, 35, 36).”

Would the cross-section analysis have more power if the two potential HLA alleles associated with the PR-E138 DRM were compared simultaneously against a reference group containing all other alleles? In the current analysis, the two potentially important alleles are essentially compared against each other (as part of the "other" category).

While combining the two HLA alleles would provide more statistical power, the issue is that one pair with RT-E138 ( with HLA-B18), already demonstrates a very strong relationship, while the other (with HLA-A24) shows none in our analysis. A pooled analysis would very likely show a relationship, but it would almost certainly be due to the RT-E138:HLA-B18’s strongly significant association “overpowering” the RT-E138:HLA-A24’s null association, and misleadingly yielding a statistically significant association overall.

It would be helpful if the frequencies in Figure 3A would be spelled out in text rather than just shown in the table. E.g. what proportion of patients had the DRM-associated HLA alleles, and what fraction of these patients subsequently acquired the DRMs?

The manuscript has now been updated to clearly explain these numbers (and percentages) within the text itself (lines 324-332):

“For RT-E138:HLA-B18, 63 (7.7%) of the 813 patients without an RT-E138 mutation were HLA-B18, among which 5 (7.9%) developed it before ART initiation, compared to the 5 (0.7%) out of the 750 with another HLA-B18 type (HR: 12.211, 95% CI: 3.523-42.318 [p value <0.001]) (Figure 3). […] Of the 150 (18.3%) of the 821 patients with HLA-B35 (initially without an HLA-B35 mutation), 3 (2.0%) developed a mutation at RT-V179, compared to only 1 (0.1%) of the 671 with another HLA-B type (HR: 16.116, 95% CI: 1.673-155.216 [p value = 0.016]).”